# Understanding the Pathogenesis of Cardiac Complications in Patients with Propionic Acidemia and Exploring Therapeutic Alternatives for Those Who Are Not Eligible or Are Waiting for Liver Transplantation

**DOI:** 10.3390/metabo13040563

**Published:** 2023-04-16

**Authors:** Evelina Maines, Michele Moretti, Nicola Vitturi, Giorgia Gugelmo, Ilaria Fasan, Livia Lenzini, Giovanni Piccoli, Vincenza Gragnaniello, Arianna Maiorana, Massimo Soffiati, Alberto Burlina, Roberto Franceschi

**Affiliations:** 1Division of Pediatrics, Santa Chiara General Hospital, APSS, 38122 Trento, Italy; 2Division of Cardiology, Santa Chiara General Hospital, APSS, 38122 Trento, Italy; 3Division of Metabolic Diseases, Department of Medicine-DIMED, University Hospital, 35128 Padova, Italy; 4Division of Clinical Nutrition, Department of Medicine-DIMED, University Hospital, 35128 Padova, Italy; 5Emergency Medicine Unit, Department of Medicine-DIMED, University Hospital, 35128 Padova, Italy; 6CIBIO, Department of Cellular, Computational and Integrative Biology, Italy & Dulbecco Telethon Institute, Università degli Studi di Trento, 38123 Trento, Italy; 7Division of Inherited Metabolic Diseases, Reference Centre Expanded Newborn Screening, Department of Women’s and Children’s Health, University Hospital, 35128 Padova, Italy; 8Division of Metabolism and Research Unit of Metabolic Biochemistry, Bambino Gesù Children’s Hospital-IRCCS, 00165 Rome, Italy

**Keywords:** propionic acidemia, cardiomyopathy, long QT syndrome, pathogenesis, review

## Abstract

The guidelines for the management of patients affected by propionic acidemia (PA) recommend standard cardiac therapy in the presence of cardiac complications. A recent revision questioned the impact of high doses of coenzyme Q10 on cardiac function in patients with cardiomyopathy (CM). Liver transplantation is a therapeutic option for several patients since it may stabilize or reverse CM. Both the patients waiting for liver transplantation and, even more, the ones not eligible for transplant programs urgently need therapies to improve cardiac function. To this aim, the identification of the pathogenetic mechanisms represents a key point. **Aims:** This review summarizes: (1) the current knowledge of the pathogenetic mechanisms underlying cardiac complications in PA and (2) the available and potential pharmacological options for the prevention or the treatment of cardiac complications in PA. To select articles, we searched the electronic database PubMed using the Mesh terms “propionic acidemia” OR “propionate” AND “cardiomyopathy” OR “Long QT syndrome”. We selected 77 studies, enlightening 12 potential disease-specific or non-disease-specific pathogenetic mechanisms, namely: impaired substrate delivery to TCA cycle and TCA dysfunction, secondary mitochondrial electron transport chain dysfunction and oxidative stress, coenzyme Q10 deficiency, metabolic reprogramming, carnitine deficiency, cardiac excitation–contraction coupling alteration, genetics, epigenetics, microRNAs, micronutrients deficiencies, renin–angiotensin–aldosterone system activation, and increased sympathetic activation. We provide a critical discussion of the related therapeutic options. Current literature supports the involvement of multiple cellular pathways in cardiac complications of PA, indicating the growing complexity of their pathophysiology. Elucidating the mechanisms responsible for such abnormalities is essential to identify therapeutic strategies going beyond the correction of the enzymatic defect rather than engaging the dysregulated mechanisms. Although these approaches are not expected to be resolutive, they may improve the quality of life and slow the disease progression. Available pharmacological options are limited and tested in small cohorts. Indeed, a multicenter approach is mandatory to strengthen the efficacy of therapeutic options.

## 1. Introduction

Propionic acidemia (PA; MIM #606054) is a rare organic aciduria due to the inherited deficiency of the mitochondrial enzyme propionyl-CoA carboxylase (PCC). PCC is a biotin-dependent enzyme active in the mitochondrial matrix that catalyzes the conversion of propionyl-CoA to methylmalonyl-CoA. On PCC converge the catabolism of four essential amino acids (isoleucine, methionine, threonine, and valine), odd-chain fatty acids, the side chain of cholesterol, and propionyl-CoA derivates generated by anaerobic bacterial fermentation. The end-product of these pathways is succinyl-CoA, a substrate of the tricarboxylic acid (TCA) cycle [1]. PCC deficiency results in a decreased availability of succinyl-CoA, together with the accumulation of propionyl-CoA and its toxic metabolites, such as methylcitrate, 3-hydroxypropionate, tiglylglycine, and propionylglycine [1].

The incidence of PA is difficult to estimate due to its rarity and varies depending on the population evaluated. Estimates of incidence in Western populations range from 1:50,000 to 1:500,000 births. In other populations, such as Saudi Arabia, the incidence may be higher, around 1 in 2000 births [2].

The genetic heterogeneity of PA leads to different clinical pictures, ranging from severe early onset to mild late onset forms [3]. Patients may present with acute or chronic symptoms at any age. Major chronic disease manifestations include neurological complications, hematologic abnormalities, hearing loss, and cardiac complications such as cardiomyopathy (CM) or acquired long QT syndrome (aLQTS) [2,3,4].

CM has been considered a complication of PA since 1993 when Massoud et al. described six cases of dilated CM (DCM) out of 19 patients with PA (age range: 13 months–8 years) [5]. A study run in a larger cohort estimated that the DCM prevalence rate is 23% (6 out of 26 patients, with a mean age of the onset of 7 years; age range: 5–11 years) [6]. A recent longitudinal observational monocentric study of 18 PA patients described CM in 7 patients (39%, age range: 3–19.5 years) [7]. Two of the patients had DCM, whilst five patients presented hypokinetic non-dilated cardiomyopathy. Finally, a case series of 10 PA patients (ages ranging between 2.5 and 20.2 years) observed reduced fractional shortening (FS) < 30% in 40% of the cases [8]. 

Although DCM is the type of CM associated more closely with PA, hypertrophic cardiomyopathy and left-ventricular non-compaction have also been diagnosed [9,10].

No correlation exists between CM and metabolic stability, degree of the phenotype, or remaining enzymatic activity [4]. CM can rapidly progress and lead to arrhythmias, heart failure (HF), and cardiogenic shock [4]. 

aLQTS has been considered a complication of PA since 1993 when Massoud et al. reported the first case [5]. The estimated prevalence is not clear, ranging from 22% to 70% of patients [8,11]. No correlation exists between the occurrence of prolonged QTc and biochemical indices [8], the number of metabolic decompensation, and the severity of the phenotype [7]. Arrhythmias may lead to severe events, including sudden cardiac arrest (SCA) [12,13]. 

The guidelines for the management of patients with PA recommend regular cardiac examinations, including electrocardiograms and echocardiograms [3,4]. If the patient presents CM or long QTc, standard cardiac therapy should be implemented together with optimized metabolic treatment and monitoring [3]. 

The more recent revision argued the clinical relevance of high doses of coenzyme Q10 (CoQ10) on cardiac function [4].

The standard of care for HF in adult patients has recently been renewed in the guidelines of the European Society of Cardiology (ESC) and the American Heart Association (AHA) [14,15]. For the pediatric age, the management of HF was defined in a consensus paper of the International Society of Heart and Lung Transplantation (ISHLT) [16]. 

Despite an improvement in the overall prognosis over the past few decades, the long-term outcome of PA patients with cardiac complications remains unsatisfactory. Liver transplantation (LT) has been proposed as a therapeutic option for selected PA patients with metabolic instability or CM. Some studies included very small numbers of patients with varying posttransplant follow-up periods and outcomes. A very recent systematic review and meta-analysis of currently available literature on LT in PA patients (including 21 studies involving 70 individuals) concluded that despite the risk of LT-related complications, LT is a safe and beneficial therapeutic option, with good patient and allograft survival rates [17]. However, the recurrence of CM after LT has been reported [18,19]. Finally, not all PA patients are eligible for liver transplant programs. 

Patients waiting for liver transplantation, as well as those not eligible for the transplant program, need therapies aiming at improving or preventing CM. Towards this goal, the identification of the pathogenetic mechanisms underlying CM and aLQTS represents a key achievement.

## 2. Aim of the Review

This review summarizes:−The current knowledge of the pathogenetic mechanisms responsible for cardiac complications in PA;−The therapeutic options for the prevention or treatment of cardiac complications in PA.

## 3. Material and Methods

To select articles, we searched the electronic database PubMed using the Mesh “propionic acidemia” OR “propionate” AND “cardiomyopathy” OR “Long QT syndrome”. 

The criteria for inclusion in the review were: (i) study population: PA patients or PA models AND CM OR aLQTS; (ii) type of study: any type; (iii) contents: studies reporting data about pathogenetic mechanisms responsible for CM or aLQTS or about treatments performed in the study population; (iv) publication date: last 20 years (2002-2022); (v) language: English.

Exclusion criteria: (i) full paper not available; (ii) studies not yet published; (iii) studies not reporting data about pathogenic mechanisms or treatments in the study population; (iv) studies reporting data about liver transplantation; (v) languages other than English.

Abstracts were reviewed, and the most relevant publications were used. Next, we screened the reference list of the selected studies.

## 4. Results

Our strategy identified a total of 6907 papers. After carefully reviewing titles, we excluded 6657 records out of scope for our analysis.

Upon the examination of the full text, we further excluded 226 studies reporting outcomes not relevant to our study. We added 53 other studies by screening the reference cited, for a total of 77 studies included in this review (Figure 1).

### 4.1. Impaired Substrate Delivery to TCA Cycle and TCA Dysfunction 

The heart needs a large amount of energy to allow both ionic equilibrium and muscle fiber contraction. Under physiological conditions, energy management can be described in three main stages: substrates reach the TCA cycle, the TCA cycle, and oxidative phosphorylation (OXPHOS) [20]. The fluxes of molecules entering the TCA are severely disturbed in PA, and the contribution of propionyl-CoA metabolism to the succinyl-CoA pool is limited [1]. The surplus of propionyl-CoA sequestrates oxaloacetate and generates methylcitrate [21], further depleting the TCA cycle [22]. Finally, the accumulation of toxic metabolites that may inhibit pyruvate dehydrogenase complex or other enzymatic steps of the TCA cycle may further jeopardize the TCA cycle [1,23] (Figure 2).

In rat hearts perfused with high concentrations of propionate, severe perturbations of energy metabolism were observed, most likely due to mitochondrial CoA trapping, inhibition of fatty acid oxidation, and increased glucose oxidation. Supplementation with L-carnitine did not resolve CoA sequestration and did not change the propionate-mediated fuel switch [24]. 

Measures to promote anabolism, a low-protein diet, laxative agents, prebiotics, and antibiotics, are commonly used in PA patients to reduce exogenous and colonic-derived propionate supply [3]. A case report described the potential benefits derived from supplementation with a daily mixture of bifidobacteria in association with antibiotic therapy to restore gut microbiota bifidobacteria population against propionate producers [25]. Nevertheless, no clinical trials have specifically investigated the role of laxative agents, prebiotics, probiotics, and antibiotics in PA patients with CM or aLQTS. 

Citric acid, an anaplerotic substrate of the TCA cycle, has been suggested as a therapeutic opportunity for PA patients [26]. Citric acid (7.5 mEq/Kg per day) was administered for 4 weeks in three patients with PA. Along with the treatment, a significant increase in urinary levels of several TCA metabolites compared to baseline values was noticed [27]. Nevertheless, there are no data on the use of citric acid in PA patients suffering from CM or aLQTS. 

### 4.2. Secondary Mitochondrial Electron Transport (mtETC) Chain Dysfunction and Oxidative Stress

OXPHOS deficiencies have been reported in the myocardium of PA patients with CM, including complex I [9], complex III [9,28], or complex IV [29]. Low complex I activity has also been observed in vivo in a genetic mouse model of PA (Pcca^−/−^) (A138T). In particular, an increase in reactive oxygen species (ROS) production was documented in heart sections, suggesting that oxidative stress may be part of the pathophysiology of CM [30].

Cardiomyocytes derived from induced pluripotent stem cells (iPSCs) of a PA patient also showed a significantly decreased OXPHOS, as suggested by the decrease in the ATP-coupled oxygen consumption ratio, maximal oxygen consumption rate, and reserve capacity compared to the controls [31]. 

These observations are in accordance with the alterations in mitochondrial OXPHOS function (low activities of several mtETC complexes) and redox homeostasis (increased ROS and oxidative stress markers) reported in PA mice tissues [30] and in samples from PA patients [28,32,33,34].

Propionyl-CoA and other metabolites can act as mitochondrial toxins impairing OXPHOS [23,35]. Interestingly, mtDNA depletion and ultrastructural mitochondrial abnormalities, leading to multiple respiratory chain deficiency [20], have been documented in PA patients [28,29,30]. In detail, electronic microscopy of myocardial biopsy identified abnormally enlarged mitochondria with atypical crista in a patient presenting a severe DCM [29].

Since the redox state in the heart stems from the balance between ROS and intrinsic antioxidant systems [36], antioxidants have been proposed as potential therapeutic compounds for PA. Rivera-Barahona et al. investigated in the Pcca^−/−^ (A138T) mouse model the impact of two compounds acting on mitochondrial red-ox metabolism: (i) mitochondrial targeted MitoQ (basically, a lipophilic cation triphenylphosphonium—TPP- linked to ubiquinone that accumulates within mitochondria following the membrane potential), predicted to ameliorate mitochondrial lipid peroxidation, and (ii) resveratrol, a natural phenol with antioxidant capability as it induces antioxidant enzymes and mitochondrial biogenesis. The oral administration of MitoQ or resveratrol decreased lipid peroxidation and induced the expression of antioxidant enzymes [37]. Notably, PA mice treated with resveratrol or MitoQ recovered normal brain natriuretic peptide (BNP) expression levels. Accordingly, antioxidants treatment (Tiron, Trolox, resveratrol, and MitoQ) lessened ROS content and increased the levels of two antioxidant enzymes (superoxide dismutase and glutathione peroxidase) in fibroblasts of PA patients [38]. These outcomes warrant further investigations of the therapeutic efficacy of antioxidants as adjuvant therapy in CM linked to PA. Nevertheless, several doubts jeopardize the use of these antioxidants, in particular, MitoQ, in clinical practice. In fact, MitoQ may elicit mitochondrial stress. MitoQ consists of a lipophilic cation (TPP coupled to ubiquinone via an alkyl chain. The alkyl chain can increase mitochondrial inner membrane permeability, thus leading to swelling and depolarization of the mitochondria [39] and a decrease in mtDNA content [40].

### 4.3. Coenzyme Q10 Deficiency 

Coenzyme Q10 (CoQ10), also known as ubiquinone, plays a key role in mtETC. CoQ10 promotes the transfer of electrons from complex I to complex III and from complex II to complex III. Furthermore, CoQ10 spares the cell membrane from lipid peroxidation and oxidative stress by inhibiting enzymes involved in ROS production [41]. 

The oxidative stress induced by HF spoils antioxidant systems, including CoQ10 [36]. In patients with HF, low plasmatic CoQ10 concentrations associate with a poorer NYHA functional class, lower left ventricular ejection fraction (LVEF), and higher plasma concentrations of amino-terminal fragments of the BNP (NT-proBNP) [42]. Furthermore, patients presenting a more dangerous HF (classes III and IV) have reduced plasmatic and myocardial levels of CoQ10, indicating that CoQ10 reduction may follow HF severity [43]. 

Baruteau et al. [29] described markedly reduced CoQ10 levels (224 pmol/mg, reference range 942–2738), atypical cristae, enlarged mitochondria, and low complex IV activity in the myocardial biopsy of a patient with PA and severe DCM. Interestingly, CoQ10 supplementation (from 1.5 to 25 mg/Kg/day) improved DCM. However, the authors did not confirm that high-dose CoQ10 supplementation restored myocardial CoQ10 levels. CoQ10 supplementation has been discussed as a potential therapeutic option to improve left ventricular (LV) function in another paper, where a fatal HF in a pediatric patient with PA was associated with low complexes I + II and I + III in the liver, suggesting the presence of a secondary CoQ10 deficiency [25]. 

Several studies have investigated the effectiveness of CoQ10 supplementation in patients (not suffering from PA) with HF [44,45]. However, the heterogeneity of the patients’ cohorts, the different designs of the studies, and the different dosages of CoQ10 used did not allow to draw conclusive results [15,43].

### 4.4. Metabolic Reprogramming

In physiological conditions, the heart utilizes large amounts of fatty acids as substrates. However, it demonstrates remarkable metabolic flexibility as it is capable of utilizing different substrates, including ketone bodies and amino acids. In the failing heart, reduced cardiac function is accompanied by overt energy metabolism perturbations and impaired metabolic flexibility. A hallmark of HF is indeed the “switch” in the substrate consumption from fatty acids toward glucose [46]. 

Using an isotope-based metabolic flux approach, Wang et al. found that the perfusion of rat hearts with propionate brought the accumulation of propionyl-CoA, mitochondrial CoA trapping, and inhibition of fatty acid oxidation with a concomitant increase in glucose oxidation [24]. These findings provided evidence that metabolic reprogramming may lead to cardiac dysfunction in PA. It is possible that mtDNA depletion, together with oxidative damage, contributes to the decline in respiratory function and to the metabolic reprogramming towards glycolysis (Warburg effect) [30].

Ketone bodies are an alternative fount of acetyl-CoA for pyruvate carboxylation and TCA cycle anaplerosis upon fatty acid or glucose metabolism impairment. Thus, Baruteau et al. [29] tested whether D,L-beta-hydroxybutyrate (D,L-BHB) can reverse the CM seen in PA. The authors reported an improvement in cardiac function in a PA patient with severe DCM treated with D,L-BHB (200 mg/Kg/day). However, given the simultaneous co-treatments with other mitochondria agents (CoQ10, riboflavin, thiamine, L-carnitine, metronidazole, vitamin D), it was impossible to retrospectively determine the real impact of each compound. However, a racemic mixture of D,L-BHB ameliorated CM in other metabolic disorders such as multiple acyl-CoA dehydrogenase deficiencies [47] and glycogen storage disease type III [48].

### 4.5. Carnitine Deficiency

In PA, the increase in the levels of propionyl-CoA esters might determine possible direct toxicity on the myocardium and, at the same time, impairment of energy metabolism secondary to the consumption of carnitine [49]. He et al. recently investigated metabolic perturbations in organs of Pcca^−/−^ (A138T) mice under a chow diet and acute administration of [13C3]propionate. They observed the largest increase in propionylcarnitine in the Pcca^−/−^ (A138T) heart highlighting the vulnerability of the heart to high circulating propionate [50].

Baruteau et al. described reduced myocardial free carnitine levels in a patient with PA and severe DCM [29], and it is known that carnitine deficiency may induce electromyocardial alterations [51]. 

In another patient with severe DCM, the plasma acylcarnitine levels measured during acute HF were not altered. However, the correlation between plasma acylcarnitine levels and muscular or myocardial levels or levels is not robust [52]. Murdach et al. [9] reported reduced levels of carnitine (both free and total) in the myocardium but normal plasma carnitine levels in an 8-year-old patient with PA who died of HF and ventricular fibrillation. In the same patient, very low levels of complexes I + III were documented both in the cardiac muscle and the skeletal muscle, and the autopsy documented cardiac hypertrophy. 

In patients with PA, supplementation of L-carnitine (at the usual dosage of 100 mg/Kg per day) is recommended to maintain its plasmatic level within the normal range, thus improving the metabolic stability of patients [4]. However, the role of plasma supplementation in restoring myocardial carnitine pools and favoring the elimination of intramyocardial derivatives of propionyl-CoA remains to be established. In rat hearts perfused with propionate, L-carnitine supplementation did not reduce mitochondrial CoA trapping and did not modify the metabolic switch induced by propionate [24]. 

### 4.6. Cardiac Excitation-Contraction Coupling Alteration

Heart mechanical contraction is triggered by an electric stimulus in the so-called excitation–contraction (E-C) coupling. Cardiomyocyte Ca^2+^ equilibrium and E-C coupling frequently become abnormal in cardiac pathologies, such as CM and arrhythmias [53]. Tamayo et al. demonstrated that cardiac dysfunction in Pcca^−/−^ (A138T) mice associated with lower systolic Ca^2+^ release ([Ca^2+^]i transients), reduction of the sarcoplasmic reticulum (SR) Ca^2+^ intake, impaired Ca^2+^ re-uptake by the SR-Ca^2+^ ATPase (SERCA2a) pump and SERCA2 oxidation [54]. In mitochondria isolated from rat hearts, as well as in heart homogenates or cardiomyocytes, the perfusion with maleic acid and PA reduced mitochondrial membrane potential, NAD(P)H content and Ca^2+^ retention capacity, and caused swelling in Ca2+-loaded mitochondria [23]. In iPSC-derived cardiomyocytes, Alonso-Barroso et al. [31] found increased levels of proteins responding to endoplasmic reticulum stress and calcium perturbations. 

Since a reduction of SERCA activity is a recognized hallmark of CM and abnormal Ca^2+^ handling increases the risk for arrhythmias [55], an altered EC coupling could contribute to the onset of cardiac complications in PA. Intriguingly, metabolites produced in PA might also directly affect other membrane channels and, eventually, cardiomyocyte membrane polarization [8,56]. 

### 4.7. Genetics

Although the possibility exists that PA patients with cardiac complications carry mutations both in genes encoding for PCC and in other genes involved in congenital LQTS or genetic CM (i.e., ion channel subunits), this has never been reported, even if whole-exome sequencing frequently supports inherited metabolic disorders diagnosis.

Moreover, QTc interval prolongation is not congenital in patients with PA and is almost absent in infants, suggesting that cardiac complications arise as a consequence of ongoing progressive defects [8]. CM incidence increases with ageing, supporting the progressive feature of cardiac involvement in PA [7]. However, it cannot be ruled out that ion-channel polymorphisms causing subclinical alterations in ion-channel function may play a role in CM [8]. 

### 4.8. Epigenetics

Post-translational modification of histones regulates gene expression and is involved in adverse cardiac remodeling [57]. Abnormal metabolites found in patients with PA may indirectly modulate the expression of genes critical for cardiac function [58]. Acetyl-CoA, as well as other metabolites such as butyryl-CoA, succinyl-CoA, and, notably, propionyl-CoA, can participate in this process [59]. 

Two pathways can potentially explain the epigenetic regulation of gene expression involved in pathological cardiac remodeling in PA: (1) the inhibition of histone deacetylases (HDACs) by propionate, resulting in increased histone acetylation [59,60,61]; (2) histone propionylation by propionyl-CoA acting as a false-substrate for histone acetyltransferases (HATs). Propionylation of lysine 14 in histone 3 (H3K14pr) promotes chromatin accessibility and stimulates transcription [62]. In vivo, PCC deletion alters global histone propionylation levels [62].

### 4.9. MicroRNAs

MicroRNAs (miRNAs) include short, non-coding, single-stranded RNAs of 20–24 nucleotides in length. miRNAs are essential players in gene expression regulation. They act post-transcriptionally on complementary mRNA sequences leading to degradation or translational repression [63]. A single miRNA can control the expression of multiple mRNAs, while each mRNA can be the target of several miRNAs. This miRNA–mRNA network governs many pivotal biological processes, such as cellular differentiation, proliferation and apoptosis processes, cellular metabolism, and mitochondrial dysfunction [64,65]. miRNA dysregulation is implicated in various cardiac pathologies such as arrhythmias, cardiac hypertrophy, and HF [66] and may trigger cardiac complications in PA [67].

Rivera-Barahona et al. identified a clear dysregulation of miRNAs and the target mRNAs expression profile in tissue samples from the Pcca^−/−^ (A138T) mouse model and in plasma samples from patients with PA [67]. Interestingly, the miRNA expression profile identified in iPSC-derived cardiomyocytes of a patient with PA showed consistent overlap with the findings in the mouse model [34]. 

More recently, a miRNA analysis was conducted in vitro and in vivo in a series of plasma samples from PA patients and two complementary models: (1) the heart gathered from Pcca^−/−^ (A138T) mice and [2] an immortalized atrial cell line, HL-1 cells, treated with propionate. In mouse myocardium, the authors reported an overt upregulation of several miRNAs involved in heart disease, for example, the miRNA-22 regulating PI3K/AKT pathway, among the molecular trigger of cardiac hypertrophy and fibrosis [68].

### 4.10. Micronutrients Deficiencies

Beyond CoQ10 deficiency, other micronutrients may relate to the mitochondrial alterations in PA. Indeed, mtETC requires adequate levels of CoQ10, zinc, copper, selenium, and iron for efficient production of ATP [43]. Noteworthy, up to 50% of HF patients lack one or more micronutrients. Patients with HF may experience micronutrient deficiency as the consequence of reduced intestinal absorption and augmented urinary excretion due to diuretics and impaired renal glomerular or tubular function, also exacerbated by oxidative or pro-inflammatory stress [43]. 

The recent ESC guidelines for the treatment of chronic HF suggest supplementation in case of nutritional deficiencies but clarify that the use of micronutrients may be limited to a specific subset of patients [14,69]. Currently, no data describe the impact of micronutrient supplementation in PA patients with CM or aLQTS.

#### 4.10.1. Selenium

Selenium deficiency is recognized among the possible causes of HF, even in the most recent ESC guidelines [14]. In fact, severe selenium deficiency has been associated with a rare form of DCM, Keshan’s disease [70], reversible by selenium supplementation [71]. 

Selenium deficiency might also contribute to CM and aLQTS in patients with PA [8]. Selenium is present in selenocysteine, an amino acid necessary for the synthesis of selenoproteins, such as the crucial antioxidant enzymes glutathione peroxidase (GPX) and thioredoxin reductase (TXNRD). The role of selenoproteins in the heart is not completely understood, but their deficiency is known to increase oxidative stress [43]. In human cardiomyocytes, the lack of selenium jeopardizes mitochondrial function and OXPHOS, and increases intracellular ROS levels, which can be corrected by restoring selenium levels [72]. 

No study has addressed selenium deficiency in PA-related CM. Nevertheless, some studies have demonstrated a reduction of cardiovascular (CV) mortality or a positive effect on myocardial performances in non-PA patients supplemented with selenium [45,73,74]. The clinical relevance of those findings is undermined by the lack of randomized clinical trials on the use of selenium supplements alone in patients with HF. A common consensus is building on the idea that only patients with selenium deficiency could benefit from selenium supplements [43,75]. 

#### 4.10.2. Iron

Iron is directly involved in mtETC complexes as it is a critical component of many enzymes involved in ATP production [43]. Iron deficiency (ID) damages the mitochondrial function and morphology of human cardiomyocytes, leading to impaired ATP production and reduced contractility and relaxation of cardiomyocytes [76]. In non-PA patients with HF and ID, impaired oxidative mitochondrial function was documented in skeletal muscle [77,78,79]. Compared with non-ID HF, ID-HF subjects are characterized by lower muscle strength, more severe phosphocreatine depletion, and higher intracellular acidosis upon exercise. This clinical picture is consistent with an early metabolic shift to anaerobic glycolysis [79].

Currently, no study has investigated ID in PA-related CM. Nevertheless, several clinical studies have documented that intravenous (iv) iron administration in non-PA patients with HF and reduced LVEF (HFrEF) brings benefits to the symptoms, quality of life, and exercise capacity [80,81,82]. The ESC guidelines recommend testing for ID for all patients with HF needs. In the case of ID, iv iron treatment should be considered [14]. The 2017 AHA/ACC guidelines made similar recommendations [15]. 

#### 4.10.3. Vitamin D

Vitamin D deficiency has been described in some cases of reversible HF associated with severe hypocalcemia [83]. However, randomized control trials reported no major improvement in clinical outcomes upon calciferol supplementation [84]. A severe vitamin D deficiency was found in a patient with PA and severe DCM [29]. No data exist on the role of vitamin D supplementation in PA patients with cardiac complications.

#### 4.10.4. Zinc

Zinc organizes the catalytic site of >300 enzymes, including the angiotensin-converting enzyme (ACE) and zinc-dependent superoxide dismutase (Cu/Zn-SOD). Zn-dependent proteases modulate angiotensin function, underlining the crucial role of zinc in the regulation of the pathways triggered by ACE [43]. 

A recent review of the literature suggests that zinc deficiency in HF may result from a reduced intake or a diminished absorption of the micronutrient, a strong inflammatory state, the hyper-activity of the renin–angiotensin–aldosterone axis, and hyperzincuria due to HF medications [85].

Although the prevalence of zinc deficiency is uncertain, serum zinc concentrations are lower in patients with HF [43,86]. Amongst patients with HF, NYHA class, age, and use of ACE inhibitors and angiotensin II receptor blockers (ARBs) negatively affect serum zinc concentrations [43,87]. 

Nevertheless, little evidence supports zinc supplementation [86,88]. Myocardial biopsies gathered from patients with malabsorption-associated CM showed that zinc supplements increased heart content of zinc with amelioration of cardiomyocyte physiology [86]. No data are available specifically for PA. 

#### 4.10.5. Thiamine

Thiamine (vitamin B1) is required for cellular energy production [89]. Thiamine deficiency is prevalent in HF patients [90]. Nevertheless, the benefits of thiamine supplementation in HF treatment have been assessed only in small groups of patients, and the evaluation of thiamine supplementation in patients with HF has shown mixed results [91]. The most recent ESC guidelines list thiamine deficiency as a possible cause of HF [14]. The ESC guidelines recommend evaluating micronutrient supplementation, including thiamine, in case of nutritional deficiencies [14]. No data are available specifically for PA.

#### 4.10.6. Riboflavin

Vitamin B2, also known as riboflavin, plays a major role in energy metabolism. Vitamin B2 is metabolized to flavin adenine dinucleotide (FAD) and flavin mononucleotide (FMN), two key electron carriers [91]. Pilot studies demonstrate that vitamin B2 deficiency is more common in the HF population. However, the most convincing results about the beneficial role of riboflavin in cardiac function came from animal models [91,92]. In humans, B-vitamin supplementation did not significantly reduce deficiency rates in patients with HF [93]. No data are available specifically for PA.

#### 4.10.7. Biotin

Biotin (vitamin B7) is a prosthetic group shared by several key enzymes involved in energy metabolism, including PCC. Consequently, biotin deficiency could have a role in CM pathophysiology. However, cardiac tissue seems to be relatively insensitive to biotin deficiency [91], and HF has not been associated with a high prevalence of biotin deficiency [91]. 

Biotin supplementation is often used in PA patients, especially before the differential diagnosis [94,95]. Nevertheless, its use in the chronic management of PA patients lacks evidence of efficacy [57]. 

### 4.11. The Renin–Angiotensin–Aldosterone System 

The renin–angiotensin–aldosterone system (RAAS) orchestrates a pivotal physiologic response to renal hypoperfusion on the basis of HFrEF. RAAS finally results in the formation of angiotensin II which is a powerful vasoconstrictor and has detrimental effects in chronic HF, eliciting vascular and cardiac remodeling and progressive myocardial fibrosis [96]. Angiotensin-converting enzyme inhibitors (ACE-i) were the first drugs that reduced mortality and morbidity in patients with HFrEF [14]. According to the ESC, AHA, and ISHLT guidelines, treatment with ACE-I is a Class I recommendation in patients with HFrEF [14,15,16]. 

In PA patients with HF, no randomized controlled trials investigated the use of ACE-I. In a monocentric observational study, Kovacevic et al. evaluated diastolic function in 18 PA patients, seven of whom progressed towards an overt LV systolic dysfunction. All patients treated with lisinopril experienced SF deterioration over time [7].

The rise in aldosterone levels, frequently observed in patients with HFrEF despite the use of ACE-I [97], is associated with the progression of myocardial hypertrophy and fibrosis. Mineralocorticoid receptor antagonists (MRA) and the angiotensin receptor–neprilysin inhibitor sacubitril/valsartan may provide benefits [14,15]. However, the use of MRA or angiotensin receptor–neprilysin inhibitor sacubitril/valsartan in PA patients has not been evaluated. 

Na^+^-glucose cotransporter-2 (SGLT2) inhibitors are glucose-lowering drugs that reduce the CV risk in type 2 diabetes mellitus. They are the new mainstay treatment for CV diseases as they significantly reduce CV mortality, hospitalizations, and improve symptoms in patients with HF. The exact mechanisms by which SGLT2-i carry out such benefits are not completely understood and might be related to sodium balance, energy homoeostasis, and mitigation of cellular stress [98]. Recently, both the FDA [99] and the EMA [100] have produced a Drug Safety Communication on the risk of diabetic ketoacidosis (DKA) in diabetic patients treated with SGLT2 inhibitors. The increased risk of DKA suggests extreme caution in the use of these drugs in PA patients. 

### 4.12. Increased Sympathetic Activation

The persistent increase in sympathetic activation in response to HF results in elevated myocardial oxygen consumption, myocardial fibrosis, and apoptosis. Beta-blockers (BB) antagonize the detrimental effects of sympathetic activation, reduce mortality and morbidity, and improve symptoms in adult patients with HFrEF [14]. The evidence supporting the use of BB in children is weaker, and a cautious and slow up-titration is always recommended [16].

Given that, in LQTS (congenital or acquired), ventricular arrhythmias are often triggered by adrenergic activation, BB plays a key role in reducing arrhythmic risk [101]. In PA patients with aLQTS or CM, no randomized controlled trials evaluated the use of BBs. In a monocentric observational study, four patients with increased QTc were treated prophylactically with BBs (two with propranolol and two with metoprolol). No patients showed ventricular arrhythmias or cardiac events [7]. Another paper described the use of propranolol in two sisters with PA and prolonged QTc [102]. 

## 5. Discussion

PA is a complex and heterogeneous disease involving multiple cellular pathways beyond the primary enzyme defect. Therefore, it is unlikely that a single mechanism may cause cardiac complications. 

Since the heart is a highly energy-demanding organ, altered energy metabolism in the myocardium is considered a major driver of pathological cardiac manifestations in PA. Indeed, cardiac involvement, in the form of CM or arrhythmia, is common in several energy metabolism disorders, including mitochondrial electron transport defects [103] and fatty acid oxidation disorders [104]. 

In addition, since cell energy is mostly obtained thanks to the mitochondria, especially via the TCA cycle and electron transport chain, pathogenetic mechanisms targeting these pathways have been extensively investigated as potential causes of cardiac disease in PA. 

Since cardiac complications are observed in vivo despite good metabolic control and without a correlation with the metabolic severity, mechanisms other than the accumulation of toxic compounds have been suggested as causative. Deficiency of nutrients, intermediates or cofactors, mitochondrial ultrastructural changes, and increased oxidative stress can all contribute to the development of cardiac complications. Intriguing new hypotheses are emerging involving miRNA signatures and epigenetic mechanisms. miRNA expression in PA is influenced by the age of the patients and by the tissue under consideration. Clearly, multiple factors regulate the molecular adaptations to PA. All these mechanisms add to the complex pattern of the phenotypes and may mediate the chronic changes associated with the natural history of the disease [64]. 

The knowledge of these pathogenetic mechanisms comes from preclinical studies, exploiting a variety of experimental models such as: [1] in vitro systems, namely patient-derived cells, immortalized cell lines, or mouse cells (treated or not treated with propionate); [2] the Pcca^−/−^ (A138T) mice, with an estimated PCC activity of 2% [105]. 

The murine model is a robust tool for studying the pathophysiology of PA, and it allows the characterization of the cardiac phenotype [30,37]. However, mouse models have some limitations, reflecting the structural and physiological differences between mouse and human hearts [106,107]. In vitro disease models based on iPSCs offer unprecedented opportunities. IPSCs can model different genetic backgrounds, originate any type of cell in the organism, and represent an almost unlimited source of biological material for physiological investigations and preclinical drug validation.

Current guidelines for the management of patients with MMA and PA do not recommend any specific treatment for CM or aLQTS in patients with PA outside of standard cardiac therapy [2,4]. However, there are accepted medications for cardiac complications in PA and medications considered potentially dangerous (Table 1). Weak evidence is reported for CoQ10 supplementation [4] (Table 2). 

No randomized controlled trials investigated the benefits of standard cardiac therapy or CoQ10 supplements in PA patients with CM. Moreover, the finding that routine ACE-I treatment does not improve SF in PA-CM [7] urges the evaluation of further treatment options.

Few collaborative efforts aimed at a better understanding of the natural history of cardiac disease in PA. The few longitudinal descriptions of these patients come principally from case reports, anecdotal experience, or observational monocentric study. The different data collection strategies, differences in treatments, and approaches among the centers, together with the retrospective nature of the analyses, reduce the relevance of the studies described above.

The strengths of this review are: it sheds light on the pathogenesis of PA-associated cardiac complications and identifies potential targets for therapy. As such, it has the potential to guide the development of novel therapeutic approaches and improve the clinical outcomes of PA patients with cardiac complications.

## 6. Conclusions and Future Directions

Current pharmacological therapeutic options for cardiac complications in PS are limited and tested only in a few individuals. Due to the limited case numbers per center, only a multicenter approach and prospective randomized controlled studies can provide strong evidence of the efficacy of treatments.

Further studies are needed to gain a better understanding of the underlying pathogenetic mechanisms. Emerging information on pathogenesis may suggest novel therapeutic approaches not directed toward the correction of the enzymatic defects but rather targeting the dysregulated mechanisms. Although these alternative approaches may not be resolutive, they may improve the quality of life and slow the disease progression. Moreover, the correction of these abnormalities may synergize with existing therapies.

## Figures and Tables

**Figure 1 metabolites-13-00563-f001:**
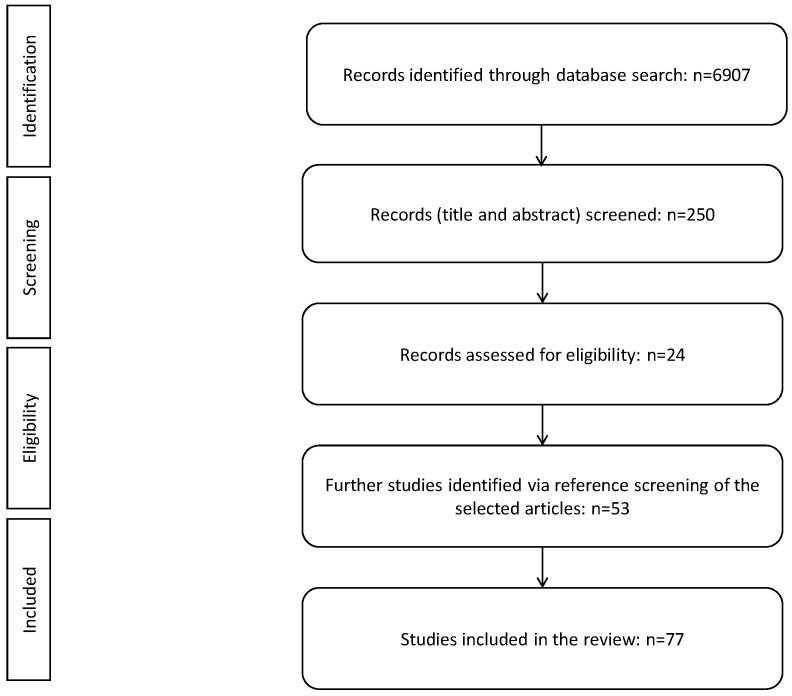
PRISMA flow diagram of study selection.

**Figure 2 metabolites-13-00563-f002:**
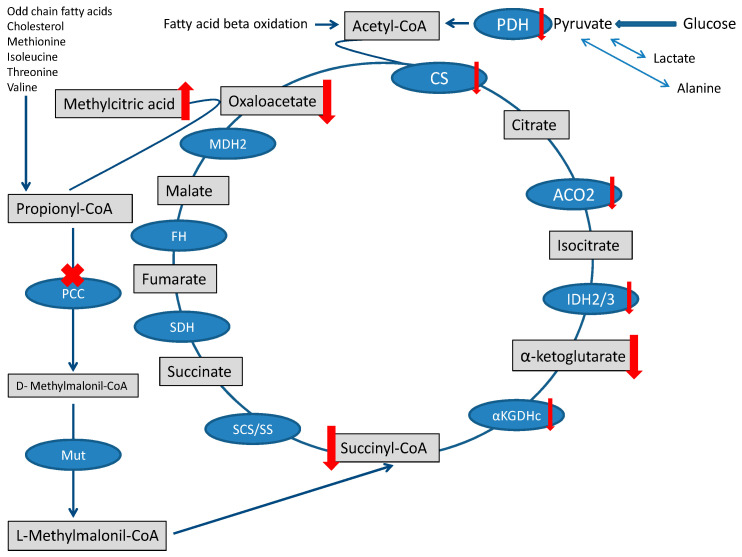
TCA cycle and Propionic acidemia. TCA is fed by the propionate pathway, glycolysis, and fatty acid oxidation (via acetyl-CoA). Methylmalonil-CoA mutase (Mut) is the last step in the propionate pathway and feeds succinyl-CoA into the TCA cycle. Intermediates and enzymes of interest are listed (enzymes are in the blue boxes). The ”toxin” methylcitric acid is also illustrated. Abbreviations: CS (citrate synthase), aconitase 2 (ACO2), isocitrate dehydrogenase 2/3 (IDH2/3), 2-αketoglutarate dehydrogenase complex (αKGDHc), succinate synthase (SS), succinyl-CoA synthetase (SCS), methylmalonyl-CoA mutase (Mut), propionyl-CoA carboxylase (PCC), succinate dehydrogenase (SDH), fumarate hydratase (FH), malate dehydrogenase 2 (MDH2), and pyruvate dehydrogenase (PDH).

**Table 1 metabolites-13-00563-t001:** Accepted medications for cardiac complications in propionic acidemia and medications considered potentially dangerous.

	Accepted	Potential Harmful	References
ACE-inhibitors (ACE-i)	X		[7]
Mineralcorticoid receptor antagonists (MRA)	X		-
Angiotensin receptor-neprilysin inhibitor sacubitril/valsartan	X		-
Na+-glucose cotransporter-2 (SGLT2) inhibitors		X	[99,100]
Beta-blockers (BBs)	X		[7,102]

**Table 2 metabolites-13-00563-t002:** Current micronutrient recommendations for patients with HF and for patients with PA.

	Heart Failure in General	References	Propionic Acidemia	References
Coenzyme Q10	Not conclusive results.	[15,43]	Weak evidence	[25,29]
Vitamin D	Lack of evidence. Vitamin D deficiency has been described in some cases of reversible HF associated with severe hypocalcemia, but routine supplementation has not proven beneficial.	[15,83]	No data	-
Thiamine	Lack of evidence. Thiamine deficiency is recognized as a cause of HF, but routine supplementation has not proven beneficial.	[14,15]	No data	-
Carnitine	Lack of evidence	[15]	Carnitine supplementation is recommended to maintain its plasmatic level within the normal range, thus improving the metabolic stability of patients	[4]
Vitamin E	Potentially harmful.	[15]	No data	-
Selenium	Selenium deficiency is recognized as a cause of HF. A common consensus is building on the idea that only patients with selenium deficiency could benefit from selenium supplements.	[14,43,75]	No data	-
Iron	Iron deficiency has been associated with clinical instability. Treatment is recommended in the case of iron depletion. Routine oral supplementation has not proven beneficial.	[14,15]	No data	-
Zinc	Routine supplementation has not proven beneficial. Little evidence supports zinc supplementation in the case of zinc deficiency.	[15,86,88]	No data	-
Riboflavin	Multivitamins have not proven beneficial.	[15]	No data	-
Biotin	Multivitamins have not proven beneficial.	[15]	Its use in the chronic management of PA patients lacks evidence of efficacy	[57]

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
