# Peer review of "Understanding the Pathogenesis of Cardiac Complications in Patients with Propionic Acidemia and Exploring Therapeutic Alternatives for Those Who Are Not Eligible or Are Waiting for Liver Transplantation"

_metabolites, 2023, doi:10.3390/metabo13040563_

Round 1

Reviewer 1 Report

Figure 1

Please give a title to figure 1 (example: PRISMA flow diagram of study selection)

Please provide a figure in order to express better the pathogenesis of propionic acidemia and TCA cycle.

Please provide a table for micronutrients deficiency in order to specify the recommendation for heart failure in general and the therapy of propionic acidemia in particular.

Please insert a table with the accepted medication for cardiac complications in propionic acidemia and the medication which is considered potential dangerous

Please complete the knowledge regarding the hepatic transplant and its utility as therapeutic option for cardiac complications in propionic acidemia.

Reviewer 2 Report

The manuscript is presenting aspects regarding one of the inborn errors of metabolism (PA); the literature review supports the involvement of multiple cellular pathways in cardiac complications of PA indicating growing complexity of their pathophysiologyand. I recommend several insertion that are into the pdf attached. Besides, I suggest to put this title: Understanding the pathogenesis of cardiac complications in patients with propionic acidemia and exploring therapeutic alternatives for those who are not eligible or are waiting for the liver transplantation

Reviewer 3 Report

This manuscript provides an overview of the pathogenesis of cardiac complications in propionic acidemia (PA) patients and summarizes the available and potential pharmacological options for the prevention or treatment of PA-associated cardiac complications. After screening 77 studies, the authors revealed 12 potential disease-specific or non-disease-specific pathogenic mechanisms and critically discussed the relevant treatment options.

While the manuscript is well-written, it is important to emphasize the significance of this review and its impact on future research. Specifically, this review sheds light on the pathogenesis of PA-associated cardiac complications and identifies potential targets for therapy. As such, it has the potential to guide the development of novel therapeutic approaches and improve the clinical outcomes of PA patients with cardiac complications.
